METHODS AND RESOURCES

# PhyloFisher: A phylogenomic package for resolving eukaryotic relationships

**Alexander K. Tice[1,2]☯, David Žihala[3]☯, Tomáš Pánek[1,3]☯, Robert E. Jones[1,2‡], Eric D. Salomaki[4‡], Serafim Nenarokov[4], Fabien Burki[5,6], Marek Eliáš[3], Laura Eme[7], Andrew J. Roger[8], Antonis Rokas[9], Xing-Xing Shen[10], Jürgen F. H. Strassert[5,11], Martin Kolísko[4,12]\*, Matthew W. Brown[1,2]\***

**1** Department of Biological Sciences, Mississippi State University, Mississippi State, Mississippi, United States of America, **2** Institute for Genomics, Biocomputing & Biotechnology, Mississippi State University, Mississippi State, Mississippi, United States of America, **3** Department of Biology and Ecology, Faculty of Science, University of Ostrava, Ostrava, Czech Republic, **4** Institute of Parasitology, Biology Centre Czech Academy of Sciences, České Budějovice, Czech Republic, **5** Department of Organismal Biology, Uppsala University, Uppsala, Sweden, **6** Science for Life Laboratory, Uppsala University, Uppsala, Sweden, **7** Unité d'Ecologie, Systématique et Evolution, CNRS, Université Paris-Saclay, Paris, France, **8** Department of Biochemistry and Molecular Biology, Dalhousie University, Halifax, Canada, **9** Department of Biological Sciences, Vanderbilt University, Nashville, Tennessee, United States of America, **10** State Key Laboratory of Rice Biology and Ministry of Agriculture Key Lab of Molecular Biology of Crop Pathogens and Insects, Institute of Insect Sciences, Zhejiang University, Hangzhou, China, **11** Leibniz Institute of Freshwater Ecology and Inland Fisheries, Ecosystem Research, Berlin, Germany, **12** Faculty of Science, University of South Bohemia, České Budějovice, Czech Republic

☯ These authors contributed equally to this work.
‡ REJ and EDS also contributed equally to this work.
\* martin.kolisko@gmail.com (MK); matthew.brown@msstate.edu (MWB)

**Data Availability Statement:** The software package is house on GitHub (https://github.com/TheBrownLab/PhyloFisher). All data associated with the figures (Figs 3, 4, and Figs A-Y in S1 Text)

## Abstract

Phylogenomic analyses of hundreds of protein-coding genes aimed at resolving phylogenetic relationships is now a common practice. However, no software currently exists that includes tools for dataset construction and subsequent analysis with diverse validation strategies to assess robustness. Furthermore, there are no publicly available high-quality curated databases designed to assess deep (>100 million years) relationships in the tree of eukaryotes. To address these issues, we developed an easy-to-use software package, PhyloFisher (https://github.com/TheBrownLab/PhyloFisher), written in Python 3. PhyloFisher includes a manually curated database of 240 protein-coding genes from 304 eukaryotic taxa covering known eukaryotic diversity, a novel tool for ortholog selection, and utilities that will perform diverse analyses required by state-of-the-art phylogenomic investigations. Through phylogenetic reconstructions of the tree of eukaryotes and of the Saccharomycetaceae clade of budding yeasts, we demonstrate the utility of the PhyloFisher workflow and the provided starting database to address phylogenetic questions across a large range of evolutionary time points for diverse groups of organisms. We also demonstrate that undetected paralogy can remain in phylogenomic "single-copy orthogroup" datasets constructed using widely accepted methods such as all vs. all BLAST searches followed by Markov Cluster Algorithm (MCL) clustering and application of automated tree pruning algorithms. Finally, we show how the PhyloFisher workflow helps detect inadvertent paralog inclusions, allowing

along with the accompanying PhyloFisher data are available from the archive https://ir.library.msstate.edu/handle/11668/19731. Additionally, the software package can be installed via Conda or PIP.

**Funding:** This project was supported primarily by the United States National Science Foundation (NSF) Division of Environmental Biology (DEB) grants 1456054 and 2100888 (http://www.nsf.gov), awarded to MWB. Support for TP's postdoctoral stay in MWB's laboratory was supported by the J.W. Fulbright Commission of Czech Republic awarded to TP. ME and MK labs are supported by the Czech Science Foundation (grants 18-18699S and 18-28103S, respectively) and the 'Centre for Research of Pathogenicity and Virulence of Parasites' (ERD funds, project no. CZ.02.1.01/0.0/0.0/16_019/0000759). ES was supported by International Mobilities of Researchers of the Biology Centre (CZ.02.2.69/0.0/0.0/16_027/0008357) and the MSCA-IF-CZ SMART (CZ.02.2.69/0.0/0.0/20_079/0017809). Research on phylogenomics in AR's lab is supported by the National Science Foundation (DEB-1442113). LE is supported by a grant from the European Research Council (ERC Starting grant 803151). FB thanks Science for Life Laboratory for supporting the work of JFHS in his laboratory, and JFHS thanks the German Research Foundation (DFG; STR1349/2-1, project # 432453260) for support. MK thanks IT4Innovations National Super Computer Center, Technical University of Ostrava, Ostrava, Czech Republic (project #Open-20-18) for providing computational resources. The funders had no role in study design, data collection and analysis, decision to publish, or preparation of the manuscript.

**Competing interests:** The authors have declared that no competing interests exist.

**Abbreviations:** BBH, best blast hit; HMM, hidden Markov model; LPP, local posterior probability; MCL, Markov Cluster Algorithm; MLBS, maximum likelihood bootstrap support; PMSF, posterior mean site frequency; RTC, relative tree certainty; SAR, Stramenopiles + Alveolata + Rhizaria; SRA, Sequence Read Archive.

the user to make more informed decisions regarding orthology assignments, leading to a more accurate final dataset.

## Introduction

Molecular phylogenetic analyses have transformed our understanding of evolutionary relationships among eukaryotes. While the use of single to a few genes provided a wealth of information when the practice was in its infancy, these small datasets lacked sufficient phylogenetic signal to resolve the deepest nodes in the eukaryotic tree of life [1,2]. In order to clarify these ancient relationships, molecular phylogenetic datasets have grown in size from single genes to hundreds of genes (i.e., tens of thousands of homologous sites) [3–5].

There are existing tools that reduce the intense labor and time needed by automating some or all parts of phylogenomic dataset construction (see, for example, [6,7]). However, as phylogenomic analyses rely on combining evolutionary signal from multiple genes into one phylogeny, it is essential to ensure that included sequences share an evolutionary history comprised only of speciation events (orthologs) rather than gene duplications (paralogs) or lateral/horizontal gene transfers (xenologs). Therefore, manual curation of orthologs is highly recommended to ensure that the dataset is free of paralogs, xenologs, or contaminants with conflicting signals that would otherwise confound phylogenetic estimation. In addition to using a high-quality set of curated orthologs, it is important that, as part of phylogenomic analyses, homogeneity of signal and potential artifacts affecting estimation is identified by exploring perturbations of the original dataset in terms of genes, sites, or taxa sampled. Despite a dramatic rise in the number of publications that include phylogenomic analyses, there is no widely accepted standard protocol for assembly of such datasets. Similarly, there are no publicly available, manually curated, starting phylogenomic databases designed for addressing deep relationships (>100 million years) in the tree of eukaryotes.

To address these problems, we have designed a protocol for phylogenomic dataset construction and data exploration and incorporated it into a software package, PhyloFisher. This publicly available software package (https://github.com/TheBrownLab/PhyloFisher) aids in the construction, maintenance, and curation of protein sequence–based phylogenomic datasets from a user-defined set of starting protein sequences. It provides tools to conduct post-dataset construction phylogenetic analyses and aids in the visualization of results. PhyloFisher also includes a manually curated starting database of 240 proteins from 304 eukaryotic taxa, representing the full breadth of known diversity in the eukaryotic tree of life (Fig 1, Table A and Fig A in S1 Text). Importantly, this database also includes identified paralogs of each of the 240 proteins from all investigated taxa, which is crucial for the identification of probable orthologs in newly added taxa. Although PhyloFisher includes this pan-eukaryotic dataset, the tool is flexible and can work with any predefined dataset consisting of protein sequences derived from eukaryotes.

To demonstrate the utility of PhyloFisher and its companion starting database to address phylogenomic questions at different depths of the eukaryotic tree of life, we performed phylogenomic reconstructions of ancient relationships in the tree of eukaryotes as a whole, as well as phylogenomic analyses of the much more recently diversified Saccharomycetaceae clade of budding yeasts using 3 different gene sets. Additionally, we show the increased accuracy of the PhyloFisher workflow over widely used methods of ortholog collection and phylogenomic dataset construction.

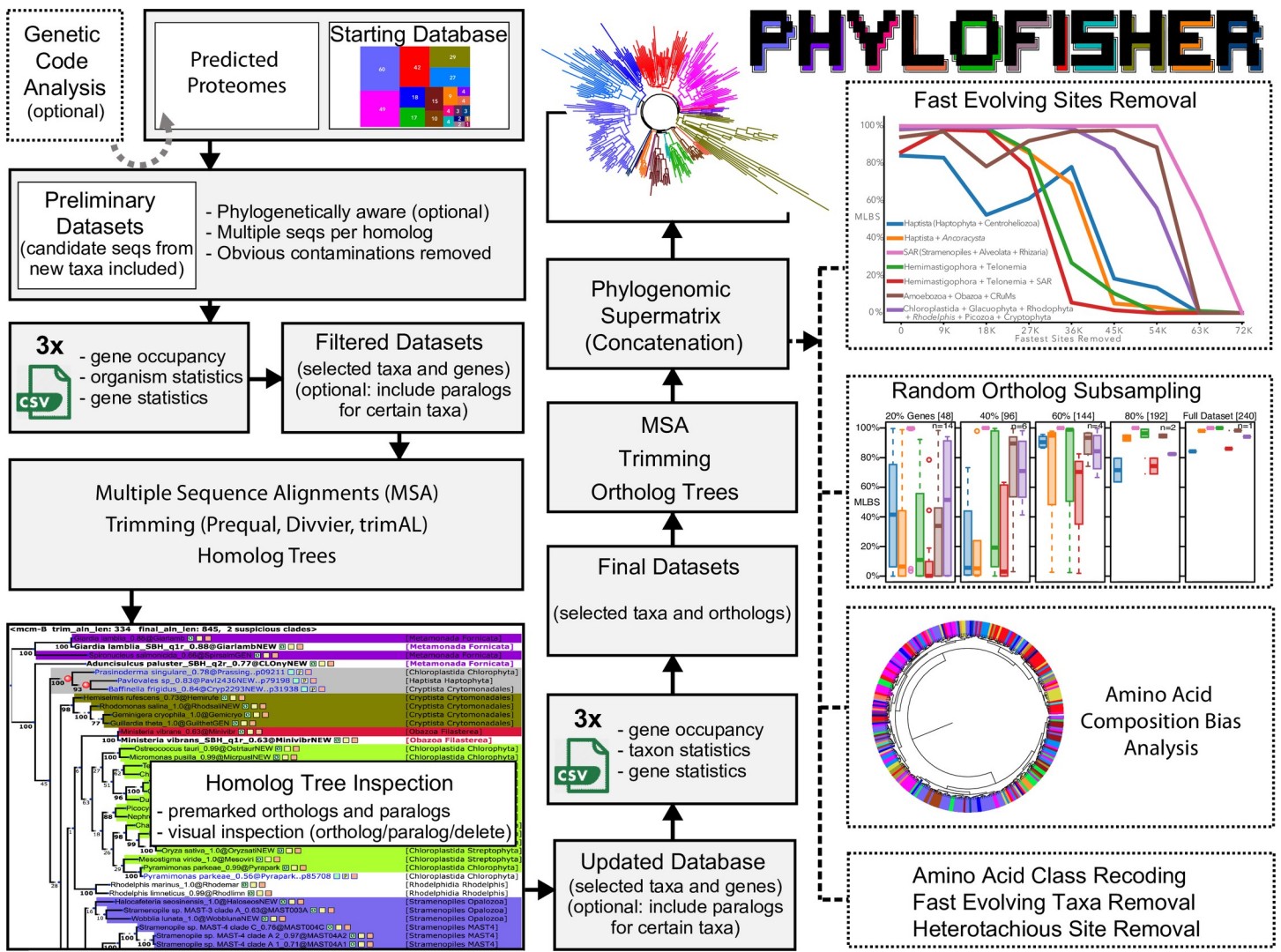

**Fig 1. Overview of the PhyloFisher workflow and package contents.** The PhyloFisher package consists of a manually curated starting database of 240 protein-coding genes and their paralogs from 304 eukaryotic taxa; a series of tools to perform the essential steps of phylogenomic dataset construction (homolog collection, single-protein tree construction, removal of paralogs and contaminants, and matrix concatenation); and many pre- and post-construction analyses necessary for a publication-quality phylogenomic study.

## Results

### The PhyloFisher protocol

The standard PhyloFisher workflow begins with the user-provided set of predicted proteins from the organism(s) they wish to add to an existing database. Alternatively, the user can create a new phylogenomic database from a set of orthologs, with or without corresponding paralogs, collected in advance prior to addition of new organisms. Users can choose between 2 fundamental protocols, one where candidate sequence selection will proceed in a novel "phylogenetically informed" way or a more traditional default manner where profile hidden Markov models (HMMs) generated from the set of orthologs in the starting database are used to collect candidate sequences from an input proteome (Fig 2; Materials and methods). Our unique "phylogenetically informed" method prioritizes putative orthologs if they branch with known

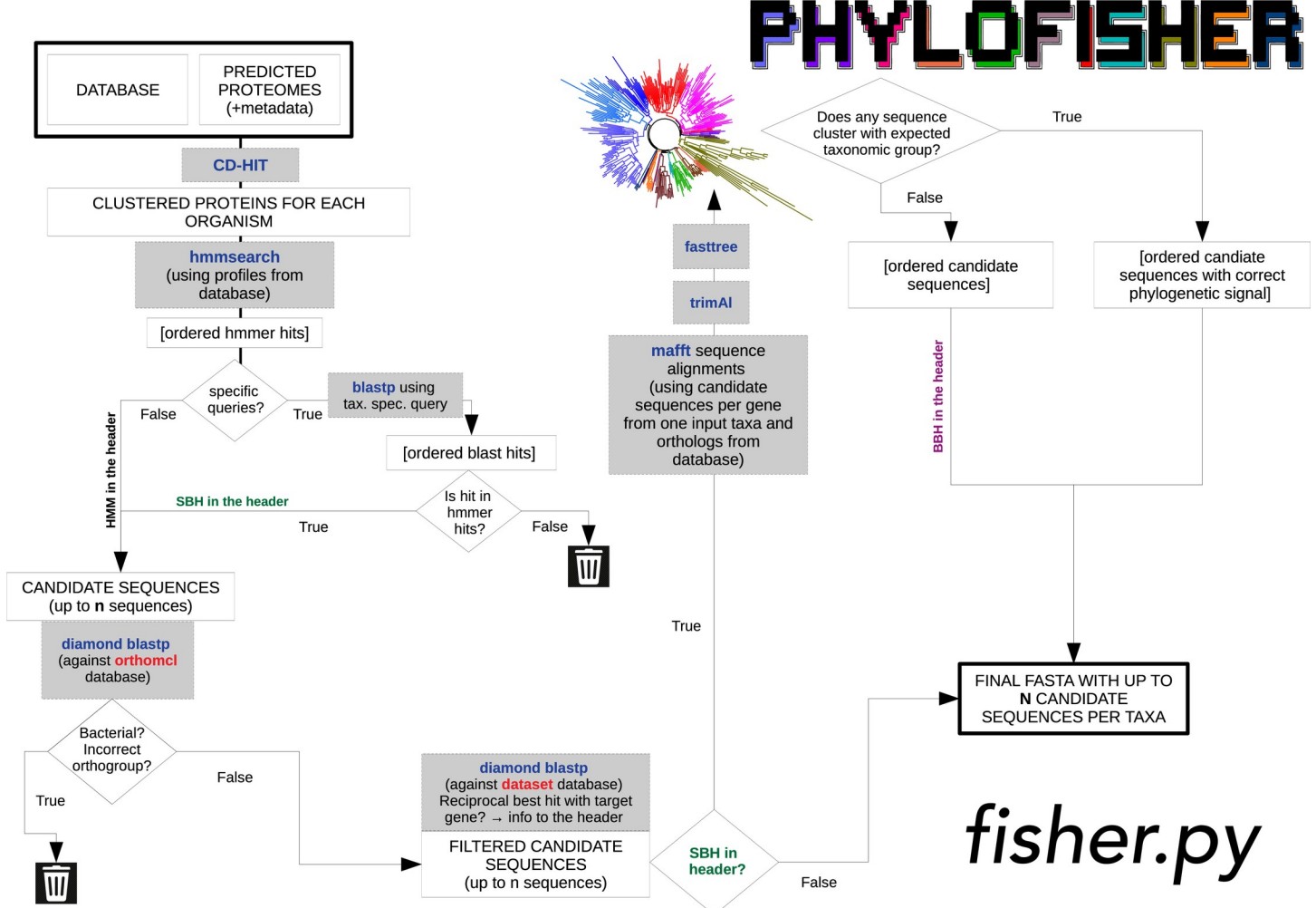

**Fig 2. Flowchart of homolog collection performed by the PhyloFisher Python script *fisher.py*.** Briefly, each predicted proteome of a new taxon to be added is processed through either a default route or a phylogenetically aware route that utilizes the manually curated orthologs from closely related taxa chosen by the user (and present in the starting database) as search queries against the proteome of the new taxon. Up to a user-defined number of collected sequences are reprioritized or eliminated based on a set of criteria designed to maximize correct demarcation of the desired ortholog and related paralogs while avoiding contaminating sequences. See Supporting information Materials and methods for a detailed description of the logic, third-party software, and associated parameters utilized.

orthologs of closely related taxa that already exist in the database (detailed in the Materials and methods section). For each protein alignment in the starting database, a user-defined number of candidate sequences from the target taxon's proteome that pass selection criteria (Fig 2; Materials and methods) is appended to the corresponding alignment of homologs, through a ranking system. The sequence receiving the highest priority by the algorithm is demarcated as the probable ortholog, and any other surviving sequences are denoted as probable paralogs. This is followed by removal of nonhomologous sites, alignment, trimming, sequence length filtering, and phylogenetic tree reconstruction of homolog datasets. The PhyloFisher workflow forces manual inspection of all single-protein trees after the addition of candidate homologs from new data to prevent the inadvertent inclusion of paralogs and sequences derived from contaminating organisms in the final phylogenomic datasets. PhyloFisher includes a graphical user interface tool (ParaSorter) designed to ease the arduous task of manual ortholog selection in large numbers of genes. ParaSorter color-codes taxon names based on their taxonomic

affiliations, highlights clades or sequences suspected to be paralogs, and appends specific sequence information to each taxon name in the phylogenetic tree. Users manually inspecting the tree can change the original designation of each sequence to either an ortholog, paralog, or "to delete" (a setting reserved for permanent removal of known contaminations) by simply clicking the corresponding box next to the sequence header (Fig 1). After the removal of "to delete" sequences, final single ortholog alignments are then generated from ortholog- or paralog-curated datasets.

Phylogenomic studies are often accompanied by additional analyses in which users manipulate or examine their dataset in various ways to reveal potentially artifactual signals. PhyloFisher is equipped with a unique set of utilities to conduct these exploratory analyses (see Table 1 for more details). These include the prediction of alternative genetic codes, removal of genes and/or taxa based on occupancy/completeness of the dataset, testing for amino acid compositional heterogeneity among sequences, removal of heterotachious and/or fast evolving sites, removal of fast evolving taxa, and supermatrix creation from randomly resampled

**Table 1. Details of phylogenomic utilities provided with the PhyloFisher package.**

| PhyloFisher utility | Description of function |
| --- | --- |
| aa_comp_calculator.py | Calculation of amino acid composition and hierarchical clustering of the data using Euclidean distances, in order to examine if amino acid composition may contribute to the groupings inferred in the phylogeny |
| astral_runner.py | Generates input files and infers a coalescent-based species tree given a set of single ortholog trees and/or bootstrap trees using ASTRAL-III [9] |
| bipartition_examiner.py | Calculates the observed occurrences of clades of interest in bootstrap trees |
| fast_site_remover.py | The fastest evolving sites are expected to be the most prone to phylogenetic signal saturation and model misspecification in phylogenomic analyses. This tool will remove the fastest evolving sites within the phylogenomic supermatrix in a stepwise fashion, leading to a user-defined set of new matrices [10] |
| fast_taxa_remover.py | Removes the fastest evolving taxa, based on tip-to-tip branch length. This tool will remove the fastest evolving taxa within the phylogenomic supermatrix in a stepwise fashion, leading to a user-defined set of new matrices with these taxa removed |
| genetic_code_examiner.py | Checks stop-to-sense and sense-to-sense codon reassignment signal in transcriptomic/genomic data |
| heterotachy.py | Within-site rate variation (heterotachy) has been shown to cause artifactual relationships in molecular phylogenetic reconstructions. This tool will remove the most heterotachious sites within a phylogenomic supermatrix in a stepwise fashion, leading to a user-defined set of new matrices |
| purge.py | Deletes taxa and/or taxonomic groups from the database permanently |
| mammal_modeler.py | Generates a MAMMaL site heterogeneous model from a user input tree and supermatrix with estimated frequencies for a user-defined number of classes using the methods described in [11] |
| random_resampler.py | Randomly subsamples the gene set included in the supermatrix into a set of new matrices. It constructs supermatrices from randomly sampled genes with user-defined options. These include sampling all genes in a random fashion within a user-defined sampling confidence interval and the percentage of subsampling a user requires per sampling step |
| rtc_binner.py | Calculates the RTC score in RAxML [12] of each single ortholog tree and bins them based on their RTC scoring into top 25%, 50%, and top 75% sets. Supermatrices are constructed from these bins of orthologs |
| SR4_class_recoder.py | Attempts to minimize phylogenetic saturation by recoding input supermatrix into the 4-character state scheme of SR4, based on amino acid binning [13] |
| taxon_collapser.py | Allows users to combine multiple operational taxonomic units into one single taxon. For example, if a user has multiple proteomes derived from single-cell libraries from a taxon, a user may decide to collapse all these libraries into a single hybrid taxon |

proteins from the starting dataset. Furthermore, PhyloFisher can consolidate multiple prote-omes from the same operational taxonomic unit to produce a single "most complete" final pro-teome. This is useful for single-cell data or to generate a single, better-sampled "hybrid taxon" representing several closely related taxa. Many of these provided utility scripts have been designed to act as "stand-alone" programs and can be used on datasets generated outside of the main PhyloFisher workflow as was done in [8].

### Reconstruction of the eukaryotic tree of life with PhyloFisher

To demonstrate the power of PhyloFisher and its accompanying database, we performed a full phylogenomic analysis of the tree of eukaryotes in IQ-TREE [14] (Fig 3, Figs A, I, and O–Q in S1 Text). We use only tools available in PhyloFisher (including the third-party software listed in S1 Text) and datasets composed of the orthologs in the provided starting database (see Materials and methods for details regarding database construction). While much of the tree of eukaryotes has been resolved over the last decade (see [15]), several key relationships remain undetermined, and some groups, often referred to as "orphan taxa," remain difficult to place on the tree. Using the PhyloFisher workflow along with our post-dataset construction analyses, we have recovered deep relationships within the eukaryote tree, which are generally consistent with results of previous phylogenomic analyses. For example, our analyses recover relation-ships such as Obazoa + Amoebozoa + CRuMs, Metamonada + Discoba, Haptophyta + Centro-heliozoa, and Ancyromonada + Malawimonada [15]. We recover a monophyletic Archaeplasatida (traditionally Glaucophyta + Chlorophyta + Rhodophyta) that includes Rho-delphidia and Picozoa and is sister to the Cryptista, although this potential relationship lacks strong statistical support (80% IQ-TREE maximum likelihood bootstrap support [MLBS]). The inclusion of Rhodelphidia and Picozoa in Archaeplastida and the group's potential sister relationship to the Cryptista has been shown recently in separate studies that used independent datasets for the phylogenomic analyses performed within [16–18].

Interestingly, our analyses also recover some novel potential relationships that deserve fur-ther investigation. For example, a clade made up of the enigmatic "orphan taxa" Telonemia and Hemimastigophora (*Hemimastix* and *Spironema*) is recovered. These 2 groups were included in phylogenomic analyses only recently, and our tree is the first to have all available data from them together in the same analysis. When the fastest evolving sites within the supermatrix are removed from the dataset using our accompanying tool *fast_site_remover.py*, this clade branches as sister to Stramenopiles + Alveolata + Rhizaria (SAR) with 95% MLBS, potentially comprising a monophyletic megagroup. Another interesting potential relationship is a group uniting the Haptophyta+Centroheliozoa clade with the "orphan" eukaryote *Ancoracysta twista* (i.e., Haptista sensu [19]). Using all the tools provided through the PhyloFisher software package, many of the abovementioned groupings are frequently recovered including when (i) fast evolving sites (Fig I in S1 Text) and heterotachious sites are removed (Figs L and O in S1 Text); (ii) genes are randomly subsampled (Fig J in S1 Text); (iii) the highest scoring relative tree certainty (RTC) ortholog alignments are concatenated (Fig Q in S1 Text); (iv) gene tree coalescent approaches are used (Fig P in 1 Text); and (v) when site heterogeneous mixture models are directly inferred from the supermatrices (Fig R in S1 Text).

### Reconstruction of the tree of Saccharomycetaceae budding yeasts and addition of *Torulaspora globosa* with PhyloFisher

To demonstrate the utility of the PhyloFisher workflow to address questions surrounding more recent divergences in the tree of eukaryotes, we reconstructed the tree of the

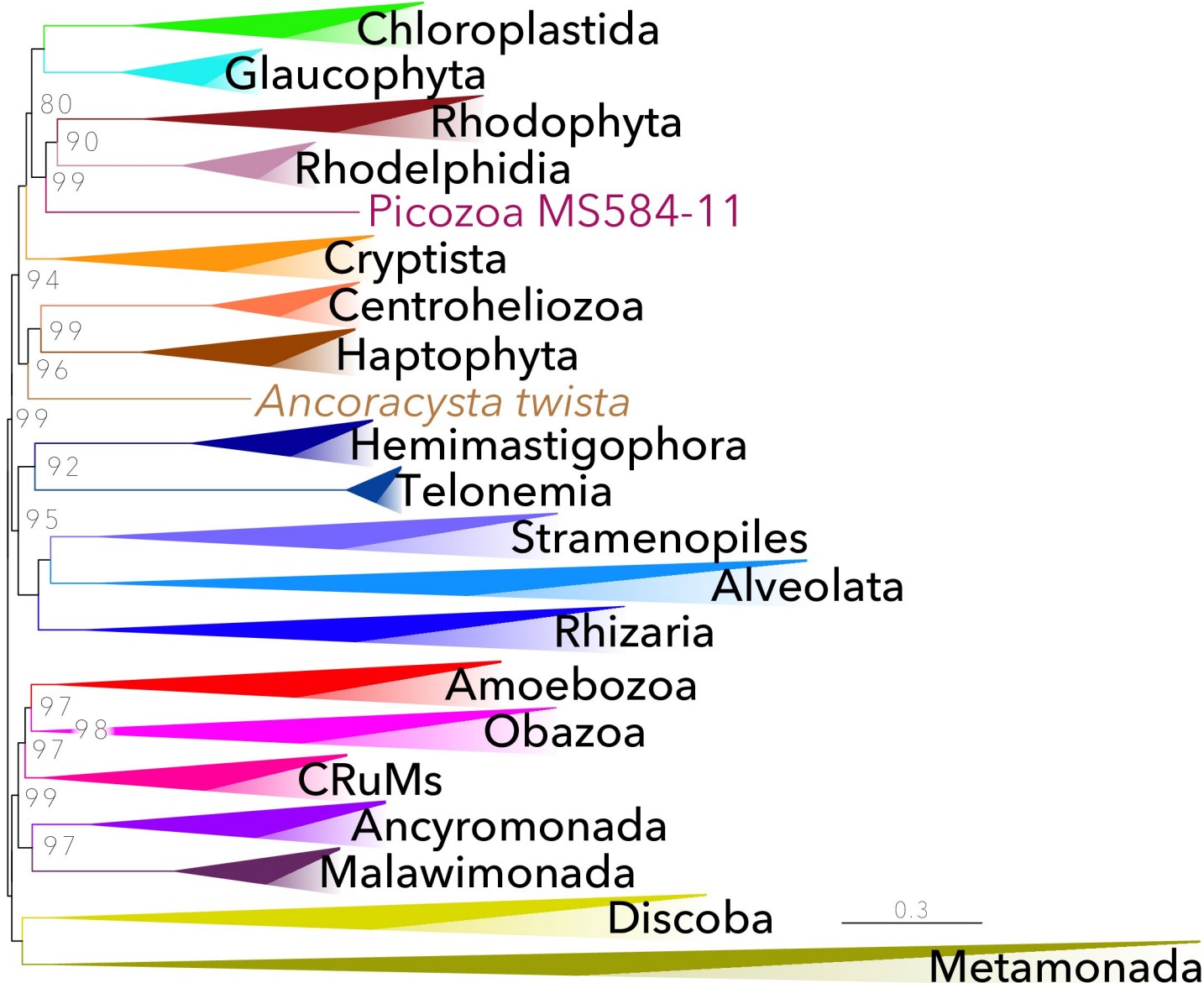

**Fig 3. Phylogenetic tree for 304 eukaryotes, inferred from 240 proteins.** The tree was inferred using ML (LG+G4+F+C60-PMSF model, with an LG+G4+F+C20 ML tree as a PMSF guide input tree) in IQ-TREE v1.6.7.1 [14]. Single-protein alignments were processed with the PhyloFisher utility *matrix_constructor.py*. See Materials and methods for details. The numbers on branches show support values from 350 ML bootstrap replicates. All nodes are fully supported (100% MLBS) unless otherwise shown. Highly supported clades of high taxonomic level have been collapsed; the full ML tree is available as Fig A in S1 Text. Taxon details are available in S1 Table. This tree was inferred from the full concatenated alignment (72,632 sites). Further detail into the methodology may be found in the Materials and methods and S1 Text. ML, maximum likelihood; MLBS, maximum likelihood bootstrap support; PMSF, posterior mean site frequency.

Saccharomycetaceae clade of budding yeasts using concatenation and coalescence-based methods. For reference, members of the Saccharomycetaceae are hypothesized to have been diverging from one another for 102 to 126 million years and exhibit a level of genetic diversity similar to that seen in flowering plants [20]. Homologs were collected from 86 budding yeast genomes, the genome of the Saccharomycetaceae *T. globosa*, and 12 other fungal taxa provided with PhyloFisher, generating a phylogenomic dataset derived from 208 of the 240 orthologs (pruned due to gene sampling in the Saccharomycetaceae representative, *Saccharomyces cere-visiae*) found within PhyloFisher's provided starting database. We also created a custom

starting database constructed of a more computationally feasible subset of 128 genes from the 1292BUSCO dataset of [20] using tools provided in PhyloFisher. These 128 genes were selected by employing the *rtc_binner.py* tool from PhyloFisher to collect the top approximately 10% RTC scoring trees of the 1292BUSCO dataset of [20], also taking into account the taxonomic coverage within the 1292BUSCO dataset (>70%), resulting in 128 genes (128BUSCO). Using the PhyloFisher workflow, we then recollected homologs of these 128 genes from the same 86 budding yeast genomes as above. We also included an additional Saccharomycetaceae taxon, *T. globosa*, to illustrate PhyloFisher's ability to add new taxa to this database. During our reanalysis of the 128 genes in the 128BUSCO dataset, we discovered the inadvertent inclusion of paralogs in 6 genes (total 63 sequences) in the data analyzed in the original study of [20] (Figs S–X in S1 Text). Additionally, during reanalysis, we were able to increase gene sampling in the original dataset for several taxa using the PhyloFisher approach (for example, see the Saccharomycodaceae clade in Fig U in S1 Text). To explore how these inaccuracies and missing data affect the structure of the tree of Saccharomycetaceae, we created a third dataset with our new decisions regarding ortholog selection and increased data sampling. Full details describing the construction of all 3 datasets can be found in the Materials and methods section.

The tree topologies for both our concatenation and coalescence analyses of the 208 ortholog subset of PhyloFisher's provided starting database are congruent with the topologies of the original 1292BUSCO dataset of [20], although MLBS support for the topology is higher (100%) using the PhyloFisher 208 ortholog dataset in our concatenated analysis than the results using the 1292BUSCO dataset (53%). However, the local posterior probability (LPP) value using gene tree coalescence is lower at this node when using the 208 ortholog dataset (Fig 4). Interestingly, the concatenation-based ML trees using both of the 128 ortholog datasets show an alternative topology to that of [20] (labeled as Shen et al. 2018—BUSCO 1292 orthologs), while the coalescence-based trees are congruent with the topology of [20], although again the LPP values are lower than those in the original study.

## Discussion

To improve and ease the accurate collection of orthologs and subsequent phylogenomic dataset construction from eukaryotic proteomes, we designed PhyloFisher as a novel workflow for these purposes and provide it to the phylogenomic community in the form of the freely available open-source software package. Included is a manually curated starting set of 240 orthologs and their related paralogs from 304 eukaryotic taxa that cover known eukaryotic diversity. However, to broaden the use of the software, we provide tools to construct starting databases from alternative sets of orthologs and paralogs that can then be harvested from new taxa and processed via the remainder of the PhyloFisher workflow. To show the potential of our newly designed software package, we reconstructed the tree of eukaryotes using our provided starting database and the tree of the Saccharomycetaceae using both our starting database and a custom database created using tools provided in PhyloFisher from a 128 gene subset of 1292BUSCO dataset used in the phylogenomic analyses of [20].

The phylogenomic tree reconstructed from the eukaryotic-wide dataset represents the most taxonomically comprehensive deep phylogeny of eukaryotes to date. Our results recover many previously known relationships as well as some potential groupings that are novel (Fig 3, Fig A in S1 Text). Although the overall branching patterns were consistently recovered in most analyses we have conducted, many of the groups are sparsely sampled, and low bootstrap support values for deepest nodes remain. For example, because the Telonemia, Hemimastigophora, and *Ancoracysta* all represent sparsely sampled lineages on these trees that can be notoriously

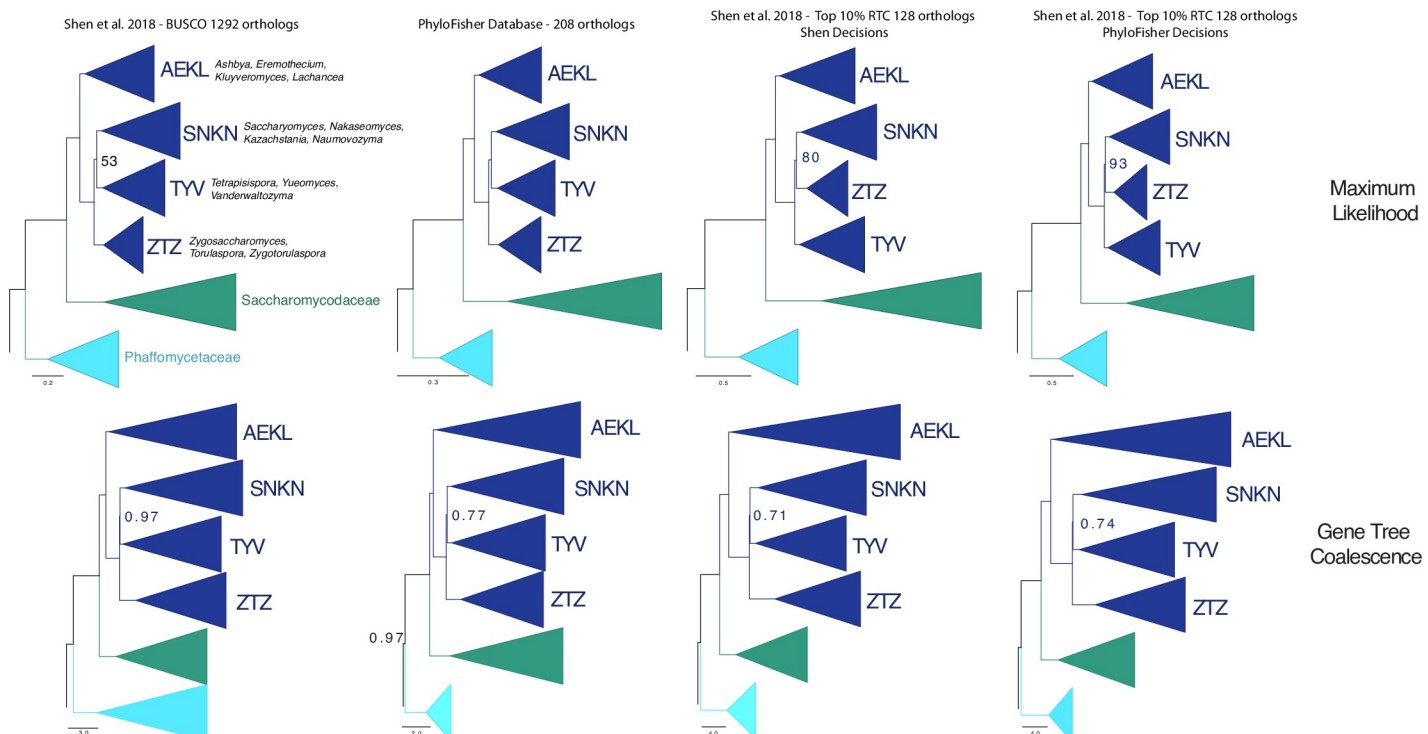

**Fig 4. Phylogenetic reconstruction of the tree of Saccharomycetaceae using 4 different datasets.** ML trees (top row) were collected from [20] in A and built using LG+G4+F+C60-PMSF model, with an LG+G4+F+C20 ML tree as a PMSF guide input tree in IQ-TREE v1.6.7.1 [36] for B, C, and D. Gene tree coalescence trees (bottom row) were collected from [20] in A and built using *astral_runner.py*, which employs ASTRAL-III [9]. The corresponding dataset a column of trees is derived from is shown across the top of the figure. Sub-clades that make up the Saccharomycetaceae are shown in dark blue (comprised of AEKL, SNKN, TYV, and ZTZ clades), while the outgroup clades of the Saccharomycodaceae and the Phaffomycetaceae are shown in dark green and cyan, respectively (labeled S and P, respectively). To the right of each Saccharomycetaceae clade is an abbreviation made up of the first letter of each genus included in the clade. Full genus names are written out to the right of the upper left ML tree. Nodes are maximally supported (100 MLBS or 1.0 LPP) unless otherwise shown. The full tree of the PhyloFisher 208 dataset is available in the Supporting information (Fig Y in S1 Text). LPP, local posterior probability; ML, maximum likelihood; MLBS, maximum likelihood bootstrap support; PMSF, posterior mean site frequency.

unstable and form artefactual groups. Thus, the foregoing relationships should be treated with caution. Notably, the exact relationships between the fully supported groups Chloroplastida + Glaucophyta, Rhodophyta + Rhodelphidia + Picozoa, and Cryptista remains unclear. Notably, the results of a recent independent study using a larger dataset [17] and another that includes novel data from diverse picozoans [18] are both congruent with our results regarding these potential relationships, although with similar or higher statistical support, respectively. Whereas the recently discovered Rhodelphidia contain a non-photosynthetic plastid and can thus fit naturally into the Archaeplastida [16], Picozoa seem to be devoid of a plastid [18,21], suggesting its secondary loss or the need to reconsider the concept of the Archaeplastida [18]. These hypothetical relationships can be tested in the PhyloFisher framework once novel species that break these branches are eventually sampled.

Our resulting yeast phylogenies using datasets derived from PhyloFisher's provided set of genes and using a custom subset of genes derived from the 1292BUSCO dataset of [20] demonstrate the potential of the PhyloFisher dataset and the workflow applied to custom datasets as tools for resolving phylogenetic questions involving a particular group of eukaryotes. During our reanalysis of the custom gene dataset used in the yeast analyses, we discovered the accidental inclusion of paralogs in the original dataset. The inclusion of paralogs in the original dataset may have led to artifactual branching patterns or low resolution at internodes in the

resulting phylogeny as has been shown to be the result of paralog inclusion [17,22]. This dataset, as well as other datasets used in the original study, was constructed via well accepted and widely used methods of all versus all BLAST searches followed by MCL clustering and subsampling of the resultant orthogroups into individual sets of orthologs via an automated tree pruning strategy. Our discovery of paralog inclusions suggests that implementation of current standard workflows can lead to the inclusion of the paralogs we discovered in our small sample (approximately 10%) of the data from the 1292BUSCO dataset of [20]. Many variables exist in this standard workflow, and subsequent studies should focus on elucidating the exact causes of ortholog misidentification during these widely used strategies for phylogenomic dataset construction. We initially considered these accidental paralog inclusions to be the likely explanation for the alternative topology with regard to the internal branching order inside the Saccharomycetaceae clade that was produced by this dataset (Fig 4). However, correcting these misassignments of paralogy only strengthened the support for the alternative topology (Fig 4), suggesting that other factors could be driving the topology produced by the 2 datasets. Further analyses of the entire original datasets used in [20] will likely be needed to resolve this contentious node in the tree of the Saccharomycetaceae.

We encourage others to explore these data, and, most importantly, add novel and unrepresented/under-sampled lineages to these datasets. A key feature of the PhyloFisher package is that even though the ortholog selection steps are manual, the software retains information from all previous curation steps. Thus, the orthologs versus paralog decisions are not permanent, rather, they are changeable for future researchers to explore and add new data. As phylogenomics is an ever-evolving science, PhyloFisher provides users the means to scrutinize these and add any new data generated. In time, we expect that by using our tool, a clearer picture of the deep relationships among eukaryotes will emerge.

We would also like to encourage educators teaching a phylogenomics course to consider PhyloFisher as their training software of choice. It is thoroughly documented and with a single installation command is "ready out of the box" for trainees to perform all standard steps in a phylogenomic study (homolog collection, data quality control, homolog tree construction, removal of paralogs and contaminants, different data sub-setting strategies, and matrix concatenation) as well as post-dataset construction analyses to test for robustness of the initial results. These attributes make PhyloFisher an ideal training tool for introducing phylogenomic analyses and concepts even if all the exact methodologies used within it are not universally agreed upon by the phylogenomics community.

In conclusion, phylogenomic analyses were historically performed by a few experts in phylogenetics and bioinformatics. More recently, phylogenomics has become a much more widely used tool in diverse biological fields as demonstrated by the rapid increase in the number of original papers that include phylogenomic trees. PhyloFisher is an easy-to-use software package, providing all the tools necessary to construct, perform quality control, and analyze large phylogenomic datasets consisting of many proteins. We show that the package and included database have the potential to help resolve deep (>100 million year) divergences in the tree of eukaryotes. PhyloFisher will also permit less bioinformatically proficient scientists to perform publication-quality phylogenomic analyses of eukaryotic lineages in a few simple steps. For more advanced users, PhyloFisher provides the flexibility to use a custom starting database, alter premade decisions in the provided database regarding ortholog selection, use alternative software other than what is provided, and change graphical aesthetics to fit their preferences. Moreover, by design, PhyloFisher encourages sharing of the resulting complete datasets with the community. We hope that this tool will facilitate the widespread use of "best practices" in phylogenomic analysis and provide long-term updating/maintenance of phylogenomic datasets.

## Materials and methods

### Fisher algorithm details (used for ortholog collection)

Ortholog selection and paralog demarcation are performed using the Python script *fisher.py* that requires 2 inputs: predicted proteome(s) to be added to the starting database and an input metadata file containing key information about them (see S1 Text and the PhyloFisher manual on the GitHub repository). To remove redundancy in the input proteome (typical when proteomes are predicted in silico from transcriptomic data), *fisher.py* invokes CD-HIT v.4.8.1 [23] with the option "-c 0.98," which clusters sequences globally at a similarity threshold of 98%. Next, hmmsearch from the software package HMMER v. 3.2.1 [24] is run on the input proteome using precomputed profile HMMs of each protein alignment in the starting database. Up to a user-defined number of sequences are collected (default = 5) that meet the significance threshold e-value < 1e -10. If no sequence meets the significance threshold, the script moves on to the next protein. If sequences are found (up to the user-defined number), they are prioritized based on the level of significance (i.e., the sequence with the most significant hit is preliminarily denoted as the most likely putative ortholog with other included sequences preliminarily denoted as putative paralogs). From here, the algorithm proceeds in either of 2 directions, depending on information given in the input metadata file. In the 2 routes outlined below, sequences can be added to or removed from the initial sequences collected by hmmsearch, as they meet or fail to meet the outlined criteria. The prioritization of a sequence can also change within the list based on the criteria outlined below. At the end of either route, if only 1 sequence remains, it is considered the putative ortholog for the input taxon. If more than 1 remains, the sequence with the highest priority in the list is considered the putative ortholog, and the rest are considered putative paralogs. All surviving sequences are added to the corresponding single-protein alignment.

### Default route

If no related species present in the starting database is/are listed in the "Blast Seed" column of the input metadata file to use as specific queries, each sequence collected from the hmmsearch run is used as the query in a search using DIAMOND v. 09.24 [25] against OrthoMCL v 5.0 [23] with the options "blastp -e 1e-10 --more-sensitive". Any sequence that has a significant hit (e-value < 1e -10) to a bacterial orthogroup in OrthoMCL is discarded along with any sequence that does not have a significant hit to the OrthoMCL orthogroup corresponding to the respective profile HMM used to retrieve the sequence. Retained sequences are again used as queries in a DIAMOND search using the parameters given above against the starting database. Query sequences whose best hit represents the conserved gene corresponding to the initial profile HMM that retrieved the query sequence is retained and added to the alignment. The remaining sequence that has the highest priority based on the initial hmmsearch is considered to be the putative ortholog, and any other surviving sequences are labeled putative paralogs.

### Phylogenetically informed route

The input metadata file gives users the option to specify taxa from the starting database whose sequences will be used as queries for blast searches against the new organism's proteome; typically, these should be closely related to the newly added taxon. Any number of species already present in the database may be chosen to serve as specific queries, but the algorithm will use them in the order provided, and, in cases outlined below, may not proceed to subsequent organisms. If organisms to be used as specific queries are listed in the input metadata file,

*fisher.py* will initially pick the sequence representing a particular orthologous gene group ("ortholog" for simplicity) from the first organism listed, if present. If absent in the starting database, *fisher.py* will sequentially check for an ortholog from the remaining taxa listed. If the orthologs from 1 or more of the subsequent taxa are found, the algorithm proceeds as outlined below. If no ortholog from all listed organisms can be found, *fisher.py* will proceed to the default route for this particular protein. If an ortholog is present in at least 1 of the listed species, then its sequence is used as the query in a BLAST [26] search against the input organism's predicted proteome. If no significant hit (e-value < 1e -10) is found, *fisher.py* will use the ortholog of the next listed species, if one is provided. If no other species is listed or if the orthologs of all listed species do not return a significant BLAST hit, the protein is skipped for the input taxon. If significant BLAST hits are found, up to a user-defined number of those (default = 5) are collected and examined further. The sequences from the original hmmsearch are reprioritized based on the level of significance of the BLAST hit. The sequence with the most significant hit becomes the priority sequence, unless it was not also collected by the initial hmmsearch: Any sequence that produces a significant BLAST hit but was not collected in the initial hmmsearch is discarded. The retained sequences are then used as queries against OrthoMCL [27]. Any sequence that has a significant hit (e-value < 1e -10) to a bacterial orthogroup in OrthoMCL is discarded along with any sequence that does not have a significant hit to the OrthoMCL orthogroup corresponding to the respective profile HMM used to retrieve the sequence. Next, all the sequences that have passed the filtering in the previous step are compared with blast against the starting database of conserved marker proteins. If the query sequence's best blast hit (BBH) is a sequence corresponding to the profile HMM that retrieved the query sequence in the initial search, the sequence is retained. If the sequence's BBH is a sequence from another alignment, the sequence is still retained, but is written out to the file "non-reciprocal_hits.txt" with a note on which protein from the starting database represented the BBH. Sequences are then added to the corresponding dataset from the starting database and aligned using MAFFT v.7.455 [28] with the parameters "--auto --reorder", trimmed with trimAl v.1.4.rev15 [29] with a gap threshold of 0.2, and subjected to phylogenetic tree reconstruction via FastTree v. 2.1.11 [30] with default parameters. The resulting tree is examined using the Python package ETE3 [31]. Sequences that branch sister to or within a clade composed of organisms with the same assigned higher taxonomy are prioritized over sequences that do not, regardless of previous criteria; higher taxonomy for taxa is derived whether from the starting database's metadata file or the input metadata file. All retained sequences are then added to their corresponding alignments, with the sequence that received the highest level of priority being denoted the putative ortholog and all others denoted as putative paralogs.

## Automated filtering, alignment, trimming, and tree construction

PhyloFisher also includes a Python script *sgt_contructor.py* to automatically filter, align, trim, length filter, and construct phylogenetic trees from all single-protein alignments. The script takes the output files from *fisher.py* (original single ortholog alignments that now contain newly added sequences selected by the fisher algorithm for input taxa as well as the previously denoted paralogs from the starting database) and removes any dashes to produce an unaligned set of sequences for nonhomologous character removal via PREQUAL v. 1.02 [32] using the default settings. Next, a length filtering step is performed to minimize the inclusion of proteins predicted from fragments of the same gene. This is common in proteomes predicted from transcriptomes. First, sequences are aligned with the program MAFFT using the settings "--globalpair --maxiterate 1000 --unalignlevel 0.6". This is followed by the assessment of

alignment error and uncertainty by the program DIVVIER v. 1.01 [33] using the options "-mincol 4 -divvygap." The resulting alignments are trimmed using BMGE v. 1.1.2 [34], with a gap rate cutoff of 0.3. Any sequence whose length is less than half the total alignment length, after BMGE filtering, is removed. After the removal of "short" sequences, files are prepared for single-protein phylogenetic tree construction by rerunning MAFFT and DIVVIER as above, followed by trimming with the program trimAl with a gap threshold of 0.01 [29]. Finally, single-protein tree reconstruction is performed using RAxML v. 8.2.12 [12], with the options "-m PROTGAMMALG4XF -f a -x 123 -N 100 -p 12345." When single gene tree construction begins, *sgt_constructor.py* will check to see how many gene trees are to be built and how many threads were provided by the user. If the number of gene trees to be built is greater than the number of threads provided, *sgt_constructor.py* will run as many jobs as it can using a single thread each. This process will continue until all gene trees have been built. If the total number of gene trees to be built is less than the number of threads available, *sgt_constructor.py* will distribute all threads available as evenly as possible.

## PhyloFisher v. 1.0 database construction

Sequence data were gathered for 304 eukaryotic taxa from various public resources. The identity of each taxon was carefully checked, resulting in numerous changes in names of the taxa (compared to the original data resource), reflecting developments in the taxonomy of the respective organisms (revisions in generic assignment, newly described species, etc.) retrieved from the literature or corrected identification of misidentified taxa. A complete list of taxa and the associated accession numbers are provided in S1 Table. When not available, predicted proteomes were obtained from transcriptome assemblies using TransDecoder-v5.5.0 (https:// github.com/TransDecoder/TransDecoder/releases). For many taxa, transcriptomes were de novo assembled from transcriptomic reads available in the Sequence Read Archive (SRA) database. First, sequencing errors were corrected in the raw reads using Rcorrector v. 1.0.1 [35] using default settings. Next, adaptor sequences and low quality bases were removed using trimmomatic v. 0.36 [36] with the following parameters: ILLUMINACLIP:2:30:10 SLIDING-WINDOW:4:5 LEADING:5 TRAILING:5 MINLEN:25. The corrected and trimmed reads were assembled using the de novo transcriptome assembly program Trinity v2.6.6 [34]. The predicted proteomes of the 304 eukaryotic taxa were used as input for *fisher.py*, collecting up to the default number of sequences ($n = 5$) for downstream analyses. These input proteomes can be found in the "proteomes" directory of the provided starting database that can be retrieved via wget from https://ir.library.msstate.edu/bitstream/handle/11668/19731/Tice_etal. PhyloFisherDatabase_v1.0_Apr.11.2021.tar.gz. Details about each proteome can be found in the metadata.tsv file located in this same directory. A preliminary starting database was constructed from a 240 gene subset of a previous phylogenomic dataset "BORDOR" developed in [37] to enable the use of sequences from diverse taxa for specific queries when appropriate. The program hmmbuild from the HMMER3 package was used to make profile HMMs for all 240 genes using the BORDOR alignments as input. Specific queries were provided for all organisms that have a well-agreed upon position in the tree of eukaryotes; otherwise, the default route in *fisher.py* was utilized. The most closely related organism with the most complete genome or transcriptome available was used as the specific query, followed by less complete data when available. After sequences had been collected by *fisher.py* for all organisms, the resulting files were subjected to the workflow described in the "Automated Filtering, Alignment, Trimming, and Single-protein Tree Construction" section. The resulting homolog trees were examined manually with the included tool "ParaSorter" to select orthologous sequences, demarcate paralogs, and remove contaminating sequences using the logic outlined in the

section "Phylogenomic dataset construction and analyses" of [8]. Finally, the aligned orthologs of the 304 public taxa (S1 Table) were used to produce the final profile HMMs shipped with PhyloFisher v. 1.0 as well as the sequences of the 240 orthologs and their paralogs that comprise the starting database.

## Construction of yeast datasets

RTC scores were calculated for all genes comprising the BUSCO 1292 dataset of [20]. The 10% of genes with highest RTC scores were collected and used as the starting orthologs to create a custom PhyloFisher database through the methodology implemented in the included utility *build_database.py*. The utility *build_database.py* takes any number of individual ortholog files (with an option to provide known paralogs) as input and generates the data necessary for the retrieval of the ortholog set from new taxa and subsequent analyses within the PhyloFisher workflow. The utility *build_database.py* aligns the provided set of orthologs using the "--auto" option of MAFFT [25] and creates profile HMMs for each gene alignment using the "hmmbuild" utility from the HMMER3 package [24], builds a DIAMOND blast database from the set of provided orthologs, and assigns an OrthoMCL orthogroup number(s) to each ortholog. OrthoMCL orthogroup numbers are assigned by using all sequences in a provided ortholog file as queries in a BLAST search against the OrthoMCL v. 5.0 database [27]. If a user-defined percentage (default = 10%) of sequences hit an OrthoMCL orthogroup with a significance threshold of evalue < 1e -10, then that orthogroup is assigned to the ortholog. More than 1 OrthoMCL orthogroup number can be assigned to ortholog. If the provided ortholog alignment is assigned "no group" or to an exclusively bacterial group in OrthoMCL, the gene cannot be used in the PhyloFisher workflow. Sequences representing putative homologs of this ortholog set were collected from all Saccharomycetaceae, Saccharomycodaceae, and Phaffomycetaceae genomes used in [20] as well as the recently sequenced genome of *T. globosa* (accession: GCF_014133895.1) via *fisher.py*'s phylogenetically informed route using default options. The putative homologs were processed using the remainder of PhyloFisher's main workflow, and final ortholog/paralog/contamination decisions were applied to the custom database described above.

During our analysis of the homolog trees produced, we discovered the inadvertent inclusion of paralogs in the original dataset of [20]. To explore the effect of these misassignments of the tree of Saccharomycetaceae, we created 2 phylogenomic datasets: the first maintaining all the original ortholog assignments of [20] and the second with the orthologs assigned through the PhyloFisher workflow.

We also collected putative homologs of the ortholog set that comprises PhyloFisher's provided starting database from all Saccharomycetaceae, Saccharomycodaceae, and Phaffomycetaceae used in [20] and *T. globosa* via *fisher.py*'s phylogenetically informed route using default options. From this, a third phylogenomic dataset was created from 208/240 orthologs that were present in at least 90% of newly added taxa and fungal taxa already present in PhyloFisher's provided database.

## Phylogenomic analyses

**Single ortholog processing.** For each dataset, orthologs were identified and collected from the output of the single-protein tree examination (*select_orthologs.py*, *select_taxa.py*, and *prep_final_dataset.py*, respectively). The individual ortholog fasta files were each processed with *matrix_constructor.py* using our default parameters, which are listed here in order as part of the *matrix_constructor.py* pipeline: (1) all gaps and any * (stops) in the ortholog fasta files are removed; (2) nonhomologous characters are removed via PREQUAL v. 1.02 [32] using the

default settings; (3) files were then aligned using MAFFT-GINSI [28] using the command "mafft --globalpair --maxiterate 1000 --unalignlevel 0.6 {numthreads} {gene}.aa.filtered > {gene}.aln"; (4) alignment files were then processed in DIVVIER [33] using the "partial" procedure with the command "divvier -partial -mincol 4 -divvygap {gene}.aln"; and (5) the partial DIVVIER alignments were then trimmed using trimAl with a gap-threshold of 80% "trimal -in {gene}.aln.partial.fas -gt 0.80 -phylip -out {gene}.gt80trimal.phy". The rationale for the use of an 80% gap threshold is provided in the trimming experiment section of the Supporting information Materials and methods.

**Concatenation-based species tree inference.** The final 4 sets of trimmed alignments were concatenated using *matrix_constructor.py* to generate 4 separate phylogenomic supermatrices. Phylogenomic trees were inferred from each matrix in IQ-TREE [14] under the site heterogenous model LG+G4+F+C60 with posterior mean site frequencies [38] inferred through LG+G4+F+C20 as input tree (Fig A in S1 Text).

**Coalescent-based species tree reconstruction.** Coalescent-based species trees (Fig 4 and Fig P in S1 Text) were inferred from the single ortholog trees and bootstrap trees for all datasets using ASTRAL-III [9]. Single ortholog trees for all 4 datasets were first inferred from the trimmed orthologs that resulted from *matrix_constructor.py*, {gene}.gt80trimal, in RAxML, using the command "*raxmlHPC-PTHREADS-AVX2 -f a -T 2 -m PROTCATLGF -p 123 -x 123 -N 100 -s {gene}.gt80trimal.phy -n {gene}.gt80trimal*" with 100 rapid bootstrap replicates, which were then used in gene tree coalescence inferences of species trees using ASTRAL-III. We provide a utility (*astral_runner.py*) as part of the PhyloFisher package to collect single ortholog trees and their respective bootstrap trees and generate input files and run ASTRAL-III.

## Supporting information

**S1 Table. List of taxa and sequence data sources.**
(XLSX)

**S1 Text. Table A**: Taxonomic composition of the PhyloFisher v. 1.0 dataset. **Fig A:** Phylogenomic tree of 304 taxa, 240 orthologs, and 72,632 amino acid sites (gt80 matrix). Supermatrix was processed as described above in the *matrix_constructor.py* methodology. The tree was built using IQ-TREE under LG+G4+F+C60+PMSF with an LG+G4+F+C20 input tree for generation of the PMSF site frequencies inferred in IQ-TREE with 350 real bootstrap replicates (MLBS). This is the uncollapsed version of the tree shown in Fig 3 of the main text (with the branches and nodes colored in the same way). MLBS values of 100% are not shown; all other values are indicated at their respective node. Data associated with this figure are available in the directory archive FigA.tgz within the data archive available from https://ir.library.msstate.edu/bitstream/handle/11668/19731/Tice_etal.PhyloFisher.DATA.tar.gz. **Fig B:** Violin plot of each gene's RTC score per method of trimming and untrimmed. Quartiles are drawn on the violin plots, overlaid with box and whisker plots. Data associated with this figure are available in the directory archive FigB.tgz within the data archive available from https://ir.library.msstate.edu/bitstream/handle/11668/19731/Tice_etal.PhyloFisher.DATA.tar.gz. **Fig C:** Box and whisker plot of the pairwise difference between each gene's RTC score per method of trimming to the untrimmed RTC score. Data associated with this figure are available in the directory archive FigC.tgz within the data archive available from https://ir.library.msstate.edu/bitstream/handle/11668/19731/Tice_etal.PhyloFisher.DATA.tar.gz. **Fig D:** Violin plot of each node in the ML tree's gene concordance factor assessed through IQ-TREE per method of trimming. Quartiles are drawn on the violin plots, overlaid with box and whisker plots. Data associated with this figure are available in the directory archive FigD.tgz within the data archive available from https://ir.library.msstate.edu/bitstream/handle/11668/19731/Tice_etal.

PhyloFisher.DATA.tar.gz. **Fig E:** Bootstrap values of nodes of interest, inferred in IQ-TREE under LG+G4+F+C60+PMSF with an LG+G4+F+C20 input tree for generation of the PMSF site frequencies inferred in IQ-TREE with 1,000 ultrafast bootstrap replicates (MLBS). This analysis highlights that different alignment trimming methods have little effect on the output tree and the bootstrap support values when a site heterogeneous model is used. Data associated with this figure are available in the directory archive FigE.tgz within the data archive available from https://ir.library.msstate.edu/bitstream/handle/11668/19731/Tice_etal.PhyloFisher. DATA.tar.gz. **Fig F:** Bootstrap values of a few selected nodes of interest displayed in Fig 3 are illustrated here, inferred in IQ-TREE under LG+G4+F+C60+PMSF with either an LG+G4+F or an LG+G4+F+C20 input tree for generation of the PMSF site frequencies inferred in IQ-TREE with 1,000 ultrafast bootstrap replicates (MLBS). Conflicting topologies with high support are found in the LG+G4+F input tree analysis, while nodes and topologies do not conflict when inferred with LG+G4+F+C20 as the input tree. Data associated with this figure are available in the directory archive FigF.tgz within the data archive available from https://ir. library.msstate.edu/bitstream/handle/11668/19731/Tice_etal.PhyloFisher.DATA.tar.gz. **Fig G:** Histogram of amino acid sites of the supermatrices generated per each trimming method. Data associated with this figure are available in the directory archive FigG.tgz within the data archive available from https://ir.library.msstate.edu/bitstream/handle/11668/19731/Tice_etal. PhyloFisher.DATA.tar.gz. **Fig H:** Fast site removal of sites from the whole dataset (the gt80 trimAl matrix of 72,632 amino acid sites). Each step has 9,000 sites, removed in a fastest to slowest stepwise manner to exhaustion. ML tree was inferred for each dataset in IQ-TREE under LG+G4+F+C60+PMSF with an LG+G4+F+C20 input tree for generation of the PMSF site frequencies inferred in IQ-TREE with 1,000 ultrafast bootstrap replicates (UFBOOT). The tree from 9,000 sites removed (9K) is shown in Fig I and represents the tree shown in Fig 3 of the main text. Data associated with this figure are available in the directory archive FigH.tgz within the data archive available from https://ir.library.msstate.edu/bitstream/handle/11668/19731/ Tice_etal.PhyloFisher.DATA.tar.gz. **Fig I:** Phylogenomic cartoon tree of 304 taxa 240 orthologs and 63,632 amino acid sites, with the top 9,000 of the fastest evolving sites in the original supermatrix removed (as indicated by fast_site_remover.py; see Fig H). The resulting supermatrix was processed as described above in the *matrix_constructor.py* methodology. IQ-TREE under LG+G4+F+C60+PMSF with an LG+G4+F+C20 input tree for generation of the PMSF site frequencies inferred in IQ-TREE with 200 real ML bootstrap replicates (MLBS). Branches and nodes are colored as shown in Fig 3 of the main text. MLBS values of 100% are not shown; all other values are indicated at their respective node. Data associated with this figure are available in the directory archive FigI.tgz within the data archive available from https://ir.library. msstate.edu/bitstream/handle/11668/19731/Tice_etal.PhyloFisher.DATA.tar.gz. **Fig J:** Random subsampling of using the *random_sample_iteration.py* utility. ("*random_sample_iteration.py -i gt80trimal.fastas/ -f phylip-relaxed -ci 0.95 -ps 20*"). Each replicate was inferred in IQ-TREE under LG+G4+F+C60+PMSF with an LG+G4+F+C20 input tree for generation of the PMSF site frequencies inferred in IQ-TREE with 1,000 ultrafast bootstrap replicates (MLBS). The support values of nodes of interest were calculated with the PhyloFisher utility *bipartition_examiner.py* and plotted in R using the boxplot function in the gplots library. Data associated with this figure are available in the directory archive FigJ.tgz within the data archive available from https://ir.library.msstate.edu/bitstream/handle/11668/19731/Tice_etal. PhyloFisher.DATA.tar.gz. **Fig K:** Hierarchical clustering of amino acid compositions of our supermatrix. Colors are depictions of taxa as labeled in Fig 3 of the main text. Data associated with this figure are available in the directory archive FigK.tgz within the data archive available from https://ir.library.msstate.edu/bitstream/handle/11668/19731/Tice_etal.PhyloFisher. DATA.tar.gz. **Fig L:** Heterotachious site removal of sites from the whole dataset (72,632 amino

acid). Step 0 and Step 1 have 3,000 sites removed (see rationale below and Fig M) and then each subsequent step has 9,000 sites removed using the greatest to least heterotachy ratio step-wise manner to exhaustion. ML tree was inferred for each dataset in IQ-TREE under LG+G4 +F+C60+PMSF with an LG+G4+F+C20 input tree for generation of the PMSF site frequencies inferred in IQ-TREE with 1,000 ultrafast bootstrap replicates (UFBOOT). Data associated with this figure are available in the directory archive FigL.tgz within the data archive available from https://ir.library.msstate.edu/bitstream/handle/11668/19731/Tice_etal.PhyloFisher.DATA.tar. gz. **Fig M:** Ratio of fast to slow taxa site rates, on a per site basis, estimated from a simulated dataset. This dataset was simulated under the LG+G4+C60+F model of evolution using our output tree under this model with our gt80 dataset. Fast/slow taxa site ratios were estimated using the *heterotachy.py* utility. The maximum observed ratio was 9.08 in simulated data. This set of ratios was used further as a null distribution of expected fast/slow ratios under this model. Data associated with this figure are available in the directory archive FigM.tgz within the data archive available from https://ir.library.msstate.edu/bitstream/handle/11668/19731/ Tice_etal.PhyloFisher.DATA.tar.gz. **Fig N:** Ratio of fast to slow taxa site rates, on a per site basis, estimated from our gt80 dataset with our output tree from this supermatrix inferred in IQ-TREE under LG+G4+F+C60+PMSF with an LG+G4+F+C20 input tree for generation of the PMSF site frequencies. Fast/slow taxa site ratios were estimated using the *heterotachy.py* utility. The null distribution as estimated from the LG+G4+C60+F simulation (Fig M) was used to calculate *p*-values from the top 3,000, 6,000, and 9,000 fast/slow ratios. Data associated with this figure are available in the directory archive FigN.tgz within the data archive available from https://ir.library.msstate.edu/bitstream/handle/11668/19731/Tice_etal.PhyloFisher. DATA.tar.gz. **Fig O:** Heterotachious site removal of 3,000 and 6,000 sites from the whole data-set (72,632 amino acid), removal of 3,000 (*p*-value = 0.0001) (left) and 6,000 (*p*-value = 0.003) (right). From these starting heterotachious removed datasets, 6,000 of the fastest sites were removed using *fast_site_remover.py*, Het3KFast6K (63,632 sites) and Het6KFast6K (60,632 sites). ML tree was inferred for each dataset in IQ-TREE under LG+G4+F+C60+PMSF with an LG+G4+F+C20 input tree for generation of the PMSF site frequencies inferred in IQ-TREE with 1,000 ultrafast bootstrap replicates (UFBOOT). UFBOOT values of 100% are not shown; all other values are indicated at their respective node. Data associated with this figure are available in the directory archive FigO.tgz within the data archive available from https://ir.library. msstate.edu/bitstream/handle/11668/19731/Tice_etal.PhyloFisher.DATA.tar.gz. **Fig P:** Coales-cent-based species tree from the 240 ortholog trees inferred in RAxML (under the PROT-CATLGF model with 100 bootstraps) using the default trimming methodology listed above in the *matrix_constructor.py* description. Tree was inferred by ASTRAL-III using the PhyloFisher utility, *astral_runner.py*. Values at nodes are ASTRAL bootstrap replicate values (BS). BS val-ues of 100% are not shown; all other values are indicated at their respective node. Data associ-ated with this figure are available in the directory archive FigP.tgz within the data archive available from https://ir.library.msstate.edu/bitstream/handle/11668/19731/Tice_etal. PhyloFisher.DATA.tar.gz. **Fig Q:** Cartoon tree of the tree generated by RTC sorted bins (top 75%, 180 orthologs, 63,750 sites) using *rtc_binner.py*. Input datasets for the *matrix_construc-tor.py* concatenation was gt80trimal single ortholog files. This tree was inferred in IQ-TREE under LG+G4+F+C60+PMSF with an LG+G4+F+C20 input tree for generation of the PMSF site frequencies inferred in IQ-TREE with 1,000 ultrafast bootstrap replicates (UFBOOT). UFBOOT values of 100% are not shown; all other values are indicated at their respective node. Data associated with this figure are available in the directory archive FigQ.tgz within the data archive available from https://ir.library.msstate.edu/bitstream/handle/11668/19731/Tice_etal. PhyloFisher.DATA.tar.gz. **Fig R:** Cartoon tree of the tree generated in IQ-TREE using the model generated by *mammal_modeler.py*. This tree was inferred in IQ-TREE under LG+G4+F

+ESmodel+PMSF (ESmodel = 60 rate classes estimated from the data inferred in MAMMaL) with an LG+G4+F+ESmodel input tree for generation of the PMSF site frequencies inferred in IQ-TREE with 1,000 ultrafast bootstrap replicates (UFBOOT). Note, this is without -bnni UFBOOT correction due to a bug in IQ-TREE v1.6.12. UFBOOT values of 100% are not shown; all other values are indicated at their respective node. Data associated with this figure are available in the directory archive FigR.tgz within the data archive available from https://ir. library.msstate.edu/bitstream/handle/11668/19731/Tice_etal.PhyloFisher.DATA.tar.gz. **Fig S:** Phylogenetic tree of ortholog EOG0934062S of the dataset from [5]. Tree was inferred using the *sgt_contructor.py* as detailed in the main text. Tree is the output from *parasorter*. Leaf names in bold are identified by the fisher.py algorithm as suggested orthologs. Leaf names not bolded are from the sequences collected as potential paralogs. The leaves with a colored background are those sequences from the dataset from [5]. Problematic paralogs are highlighted with red arrows, and the corrected replacement identified by PhyloFisher are highlighted by blue arrows. A downloadable figure and data associated with this figure are available in the directory archive FigS-X.tgz within the data archive available from https://ir.library.msstate. edu/bitstream/handle/11668/19731/Tice_etal.PhyloFisher.DATA.tar.gz. **Fig T:** Phylogenetic tree of ortholog EOG093409ME of the dataset from [5]. Tree was inferred using the *sgt_contructor.py* as detailed in the main text. Tree is the output from *parasorter*. Leaf names in bold are identified by the fisher.py algorithm as suggested orthologs. Leaf names not bolded are from the sequences collected as potential paralogs. The leaves with a colored background are those sequences from the dataset from [5]. Problematic paralogs are highlighted with red arrows, and the corrected replacement identified by PhyloFisher are highlighted by blue arrows. A downloadable figure and data associated with this figure are available in the directory archive FigS-X.tgz within the data archive available from https://ir.library.msstate.edu/ bitstream/handle/11668/19731/Tice_etal.PhyloFisher.DATA.tar.gz. **Fig U:** Phylogenetic tree of ortholog EOG093407UY of the dataset from [5]. Tree was inferred using the *sgt_contructor. py* as detailed in the main text. Tree is the output from *parasorter*. Leaf names in bold are identified by the fisher.py algorithm as suggested orthologs. Leaf names not bolded are from the sequences collected as potential paralogs. The leaves with a colored background are those sequences from the dataset from [5]. Problematic paralogs are highlighted with red arrows, and the corrected replacement identified by PhyloFisher are highlighted by blue arrows. A downloadable figure and data associated with this figure are available in the directory archive FigS-X.tgz within the data archive available from https://ir.library.msstate.edu/bitstream/ handle/11668/19731/Tice_etal.PhyloFisher.DATA.tar.gz. **Fig V:** Phylogenetic tree of ortholog EOG093403TH of the dataset from [5]. Tree was inferred using the *sgt_contructor.py* as detailed in the main text. Tree is the output from *parasorter*. Leaf names in bold are identified by the fisher.py algorithm as suggested orthologs. Leaf names not bolded are from the sequences collected as potential paralogs. The leaves with a colored background are those sequences from the dataset from [5]. Problematic paralogs are highlighted with red arrows, and the corrected replacement identified by PhyloFisher are highlighted by blue arrows. A downloadable figure and data associated with this figure are available in the directory archive FigS-X.tgz within the data archive available from https://ir.library.msstate.edu/bitstream/ handle/11668/19731/Tice_etal.PhyloFisher.DATA.tar.gz. **Fig W:** Phylogenetic tree of ortholog EOG09340RBX of the dataset from [5]. Tree was inferred using the *sgt_contructor.py* as detailed in the main text. Tree is the output from *parasorter*. Leaf names in bold are identified by the fisher.py algorithm as suggested orthologs. Leaf names not bolded are from the sequences collected as potential paralogs. The leaves with a colored background are those sequences from the dataset from [5]. Problematic paralogs are highlighted with red arrow. A downloadable figure and data associated with this figure are available in the directory archive

FigS-X.tgz within the data archive available from https://ir.library.msstate.edu/bitstream/handle/11668/19731/Tice_etal.PhyloFisher.DATA.tar.gz. **Fig X:** Phylogenetic tree of ortholog EOG093400WO of the dataset from [5]. Tree was inferred using the *sgt_contructor.py* as detailed in the main text. Tree is the output from *parasorter*. Leaf names in bold are identified by the fisher.py algorithm as suggested orthologs. Leaf names not bolded are from the sequences collected as potential paralogs. The leaves with a colored background are those sequences from the dataset from [5]. Problematic paralogs are highlighted with red arrows, and the corrected replacement identified by PhyloFisher are highlighted by blue arrows. A downloadable figure and data associated with this figure are available in the directory archive FigS-X.tgz within the data archive available from https://ir.library.msstate.edu/bitstream/handle/11668/19731/Tice_etal.PhyloFisher.DATA.tar.gz. **Fig Y:** Phylogenetic reconstruction of the tree of Saccharomycetaceae using the PhyloFisher 208 dataset. ML tree built using (LG+G4+F+C60-PMSF model, with an LG+G4+F+C20 ML tree as a PMSF guide input tree) in IQ-TREE v1.6.7.1 [1]. Sub-clades that make up the Saccharomycetaceae are shown in dark blue, while the outgroup clades of the Saccharomycodaceae and the Phaffomycetaceae are shown in dark green and cyan. Nodes are maximally supported (100 MLBS) unless otherwise shown. Data associated with this figure are available in the directory archive FigY.tgz within the data archive available from https://ir.library.msstate.edu/bitstream/handle/11668/19731/Tice_etal.PhyloFisher.DATA.tar.gz. ML, maximum likelihood; MLBS, maximum likelihood bootstrap support; PMSF, posterior mean site frequency; RTC, relative tree certainty. (DOCX)

## Acknowledgments

PhyloFisher was named during a fly fishing trip on the White River in Arkansas with AKT, TP, DŽ, and MWB; we thank Patrick Brown for lending his cabin. We thank Prof. Edward Susko at Dalhousie University for allowing us to distribute the programs DIST_EST and MAMMaL with our PhyloFisher package. We would like to thank Kristina Terpis for her advice on fine-scale taxonomy of the Stramenopiles. We are also grateful to Jolien van Hooff and Nicholas Fry for acting as beta testers.

## Author Contributions

**Conceptualization:** Alexander K. Tice, David Žihala, Tomáš Pánek, Robert E. Jones, Eric D. Salomaki, Fabien Burki, Marek Eliáš, Laura Eme, Andrew J. Roger, Martin Kolísko, Matthew W. Brown.

**Data curation:** Tomáš Pánek, Antonis Rokas, Matthew W. Brown.

**Formal analysis:** Alexander K. Tice, Tomáš Pánek, Robert E. Jones, Matthew W. Brown.

**Funding acquisition:** Matthew W. Brown.

**Investigation:** Alexander K. Tice, David Žihala, Tomáš Pánek, Robert E. Jones, Marek Eliáš, Andrew J. Roger, Xing-Xing Shen, Martin Kolísko, Matthew W. Brown.

**Methodology:** Alexander K. Tice, David Žihala, Tomáš Pánek, Robert E. Jones, Eric D. Salomaki, Serafim Nenarokov, Fabien Burki, Marek Eliáš, Andrew J. Roger, Antonis Rokas, Xing-Xing Shen, Jürgen F. H. Strassert, Martin Kolísko, Matthew W. Brown.

**Project administration:** Matthew W. Brown.

**Resources:** Marek Eliáš, Andrew J. Roger, Antonis Rokas, Xing-Xing Shen, Martin Kolísko.

**Software:** Alexander K. Tice, David Žihala, Tomáš Pánek, Robert E. Jones, Eric D. Salomaki, Serafim Nenarokov, Martin Kolísko.

**Supervision:** Marek Eliáš, Matthew W. Brown.

**Validation:** David Žihala, Tomáš Pánek, Eric D. Salomaki, Antonis Rokas, Xing-Xing Shen, Matthew W. Brown.

**Visualization:** Alexander K. Tice, David Žihala, Tomáš Pánek, Robert E. Jones, Serafim Nenarokov, Martin Kolísko, Matthew W. Brown.

**Writing – original draft:** Alexander K. Tice, Tomáš Pánek, Matthew W. Brown.

**Writing – review & editing:** Alexander K. Tice, David Žihala, Tomáš Pánek, Robert E. Jones, Eric D. Salomaki, Serafim Nenarokov, Fabien Burki, Marek Eliáš, Laura Eme, Andrew J. Roger, Antonis Rokas, Xing-Xing Shen, Jürgen F. H. Strassert, Martin Kolísko, Matthew W. Brown.

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
