## [Decision Letter · Decision Letter 0]

7 Oct 2020

Dear Matt,

Thank you very much for submitting your manuscript "PhyloFisher: A phylogenomic package for resolving deep eukaryotic relationships" for consideration as a Community Page at PLOS Biology. Your manuscript has been evaluated by the PLOS Biology editors, and by three independent reviewers.

IMPORTANT: Many thanks for your patience while we discussed the reviews. We took a little extra time as we became concerned about whether a Community Page is the right format for this. I'd wondered this when you first approached us before submission, and we went back and forth between Community Page and a Methods and Resources paper. Recently we have started to refocus our Community Pages (and other "magazine section" formats) to make them shorter and of broader appeal. It is also clear from the reviewers (and especially reviewers #1 and #2) feel that the paper needs to be expanded somewhat to attend to their concerns.

You'll see that reviewers #1 and #2 are pushing for the authors to demonstrate the pipeline being put through its paces on shallower trees, in order to broaden the appeal and utility of the method. Each of these reviewers had some additional, more minor concerns. Most of reviewer #3’s requests relate to their inability to test the pipeline and the paucity of documentation on Github, rather than the paper per se.

We took the liberty of discussing the rationale of changing the paper to a Methods paper with an Academic Editor with relevant expertise, and they were supportive. We therefore ask that you convert your article (on re-submission) to a Methods and Resources paper (do check the article processing charges: https://plos.org/publish/fees/); this would allow you to expand your proof-of-principle to address the reviewers' concerns, and you would also have the option of shunting some of the rather large Supplementary Info into the main article, giving it a more conventional structure.

In light of the reviews (below), we will not be able to accept the current version of the manuscript, but we would welcome re-submission of a much-revised version that takes into account the reviewers' comments. We cannot make any decision about publication until we have seen the revised manuscript and your response to the reviewers' comments. Your revised manuscript is also likely to be sent for further evaluation by the reviewers.

We expect to receive your revised manuscript within 3 months. 

**IMPORTANT - SUBMITTING YOUR REVISION**

*Re-submission Checklist*

*Published Peer Review*

*PLOS Data Policy*

*Blot and Gel Data Policy*

Sincerely,

Roli

Senior Editor,

rroberts@plos.org,

PLOS Biology

REVIEWERS' COMMENTS:

IMPORTANT:

Please change your article type to "Methods and Resources" when re-submitting.

Reviewer #1:

The manuscript entitled "PhyloFisher: A phylogenomic package for resolving deep eukaryotic relationships" presents the software PhyloFisher that aims to facilitate phylogenomics analyses using the state of the art implementations. Such software in general comes in handy, espacially for those who are new in the field and have not their main research focus on resolving phylogenies. The software is scalable and different post-phylogeny analyses are implemented. It is helpful to combine the knowledge gained in the past years of phylogenomic, large scale analyses and make them available to a broader user spectrum. I get the feeling that this software can also be used well in teaching, which is not mentioned in the manuscript. Overall the manuscript is very well written and understandable for the broader community the software is aimed for to be used. Therfore, I support the publication of the manuscript after some changes and add-ons have been performed.

My main advice is two apply PhyloFisher to a different area of the tree of life to show the full potential of the software. The single example provided is addressing a very deep node. I suggest to add an additional analysis for a much more internal node within plants, animals or such. Also, it would be good to show PhyloFishers performance on the gene selection, orthology determination of very new datasets of such internal branched. As it is known, there are many, lineage specific genes that are informative to resolve phylogenetic relationships that are not part of the curated set of genes the authors have implemented. I guess this should be possible within revision time.

The main Figure 1 can be better organized. It is quite compressed, the use of fonts is not easy to grab and a better design would be more helpful. Furthermore, the colors used are not all discriminable by color-blind readers, so please use different color sets. This BTW is true for ALL figures incl. the Supplementary Material. 

Reviewer #2:

Tice et al. describe a pipeline for building and curating large multi-gene phylogenetic datasets for inferring the phylogeny of eukaryotes. The pipeline and reference dataset are likely to be used by members of this community. The scripts are well described and feature-rich. It seems that the main purpose of the pipeline is for adding species to their database of 300 eukaryotes and 240 genes. They have used this dataset to successfully infer the tree of eukaryotes, with a focus on inferring deep splits in the tree. Researchers who wish to do this will find the approach described here very appealing. It appears that one could sequence the transcriptome or genome of a eukaryote and fairly easily place it in their tree of eukaryotes without having to master every step of performing a phylogenomic analysis, which for most people involves lots of scripting and mastering of dozens of software programs. Thus, this set of scripts appears to make this endeavor more accessible to the broader eukaryotic phylogenomics community.

It is less clear to me how useful this will be for someone who works on a particular group of eukaryotes. For example, the myBaits angiosperm 353 kit has found widespread use by the flowering plant community. Could they create a database of these orthologs and have a parallel "angiosperm" version of the pipeline? Is it flexible in this way, or would that be a lot of work? Could this easily be adopted for vertebrates? Green algae? Beetles?

This paper emphasizes manual curation of phylogenomic datasets, referring to manual intervention and inspection as "absolutely required" (line 64). I'm not sure I agree, but one advantage of their pipeline is that it reportedly logs the manual curation steps, which is important for reproducibility, which is a major problem in phylogenomics. Anyone who's carried out these types of analyses can recognize a Methods section where it was clear that the investigator made lots of undocumented decisions.

I think this manuscript somewhat trivializes the tree-pruning algorithms of Yang and Smith (reference 6 in this manuscript). That pipeline does not have all of the features of this one, to be sure, but it semi-automates many of the same steps, and it follows fixed algorithms to identify and prune paralogs, giving orthologs for gene-tree and then species-tree inference. This was probably one of the biggest advances in the field for de novo phylogenomics. This approach differs in that it is not de novo, rather one might use the approach of Yang and Smith to obtain a set of orthologs de novo, create a database of these, then adopt parts of the pipeline described here to complete the analysis or for future studies. If I have mischaracterized this, then I apologize, but I'd encourage you to clarify the manuscript. This is not for de novo phylogenomics, correct? In a related question, how independent are the various scripts in the pipeline, i.e., could one use just the scripts for setting up their Astral run or stripping high-rate sites, or would this throw a ton of errors because they haven't used the entire pipeline from step 1? I hope it's the former because some of these scripts will be very useful.

I mentioned above that I disagree somewhat that manual intervention is "absolutely required." I would also argue that (my understanding of) their description of a "phylogenetically informed/aware" approach to ortholog selection makes it sounds much more sophisticated than it actually is. "Phylogenetically informed" means that the user has identified a close relative in the reference dataset, so the HMM or BLAST searches are tailored accordingly. This is related to my friendly disagreement about manual curation as an "absolute" requirement (again, their language). In many of the papers on eukaryote phylogeny, some newly discovered or newly cultured taxon is sequenced, added to a tree, and there's a big story about it being sister to one lineage or another. These trees have dozens or hundreds of taxa, like the one in the present manuscript, so these are often deep splitting lone branches on the tree, the kind of branch that is often (and wrongfully) described as basal. So what does manual inspection of the gene trees do for you in this example? Sometimes an orthogroup includes deep-branching out-paralogs, and oftentimes single gene trees have abysmally low bootstrap (or other) support. When dealing with an organism whose phylogenetic placement is truly unknown and with hundreds of gene trees with low or variable support, is it really best practice to have the user manually selecting which sequences are orthologs and which are paralogs? I'd rather have an approach like Yang and Smith make those unbiased determinations for me. I'm not arguing here that Yang and Smith is perfect? But manual determination of paralogs for deep nodes in the eukaryotic phylogeny seems potentially fraught.

This was a well-written manuscript by a group that has clearly thought about these issues and has the benefit of experience. I think the protist and eukaryote phylogeny communities are likely to adopt these pipelines for their work, making them potentially very impactful. If the answers to all of my outstanding questions turn out to have favorable answers, some or perhaps all of the scripts will find even wider use.

Reviewer #3:

PhyloFisher

Phylofisher is a set of scripts, database and tools to generate standardised phylogenies to try and resolve interesting or unknown phylogenetic relationships in the eToL.

The author's identify the need for these tools, as there are a multitude of published trees looking at similar datasets but note that they often show differening results (and interpretations) due to the differences in their curation and methods used. They go on to create a database and set of tools in order to help with this, and also test their tool on several contentious evolutionary relationships serving as a proxy to dtermine their tool use and pipeline decisions.

The author's built a curated datsaet of 240 proteins from 304 diverse taxa from the eToL ("PhyloFisher_Proteome_Data.tgz" and detailed philosophically from Ln171++) - however, this is not currently located as a download in the supplementary for reviewers, and on contacting the journal editor they did not have a copy either. This is not particularly useful. It is also not part of the conda install, and not available on their github.

For publication this should be made available, open and placed somwhere like figshare or another repository, versioned and archived with a DOI. It is an important part of the pipeline (unless you create your own DB, which I feel is not adequately explained either), and without it the claims surrounding phylogenetic relationships further in the paper cannot be repeated or tested. Also to be able to test the tool for review it should have been accessible. This seems to be somewhat of a missed opportunity and largely reflects the review decision in my opinion.

The tools/scripts themselves are available at github, and can also be installed with modern package manager tools such as conda and pip, which is most important for a modern tool that encompasses and relies on many dependenicies - so, that is good to see. Installation for this review used miniconda on ubuntu linux, which seemed to work fine. There are a few issues with some scripts, which are noted below…

The methodology of the reconstruction of the eToL and others is quite well explained but is somewhat burried in the supplementary (ln215++), it would be good to see this in a more user friendly format on the GitHub repo (e.g. as a wiki or similar) as currently there is rather little on the functioning of the scripts located there. Also a walkthrough and some example datasets of the best use cases would be extremely valuable for users who want to learn and get started quickly. A tool that is advertised as "easy-to-use" really needs something like this accompanying it. Ln 69 mentiones a "manual", but where is this? There are a great many accessory scripts included in this tool and they should be mentioned and explained with reference to their use, expected inputs and outputs. e.g. Fig 1. Although a large image with a lot of information displayed, many of the boxes/panels are represented by scripts within the phylofisher package, therfore it would be quite useful for the names of those scripts to appear in the boxes that represent them as a hint for potential users.

Further, Ln90 talks about an "input metadata file" but this file and the format it expects are not detailed anywhere (what inputs are required, what format should they take). A version is included in the github repo but refers to hard coded details not present in the repo itself or values are left as "XX" without reference to what values they could take, so the example is not particularly useful as a starting point. There does not have to be a complete walkthough of how to assemble the same dataset the author's produced for the paper, although that would be excellent and reflect the open nature and quality of PloS (the genomes/transcriptomes are detailed in supplementary but perhaps they should be offered as a static tarball of assemblies etc), but a smaller and slightly contrived example with a small set of easily accessible Euk genomes would suffice - this should detail the full process, all commands needed, even those outside of the PhyloFisher package to show what is required to start...

Similarly, a walkthrough on the steps to create a custome DB (if not using the one referenced in the paper) would be extremely valuable, there is some barebones of this on their github already but it is not overly clear on any prior steps, and it is also unfinished...

Running "forge.py" and "fisher.py" without any arguments throws a python error, they should at least display the help menu instead.

The paper also mentions a tool "ParaSorter" (e.g. Ln108) which also appears in the supplementary, but I do not see it as an installation candidate or any code within the group's github anywhere. Or at least it is non-obvious to me. It sounds cool! So, where is it? Also, I think it probably deserves more than one runaway sentence...as it could be a useful tool outside of PhyloFisher in-and-of itself. It too should be made available as part of the review and eventual publication/repository. Apologies if I have missed it, but I went through the code where I think it should be and there's nothing there.

Experimentally, the methodologies employed to produce an alignment and the order of the specific tools used to then produce a phylogeny is sound and I have no concern over the methodology here. The selection of trimming tool is nicely corroborated with the use of RTC scores for genes trees per different settings across trimming programs. Indeed, many tools are tested in this manner throughout the script. Nonetheless, I think that it would help, e.g. those coming to a tool advertised as "easy to use", if the choices could have a little more of the author's thought processes/philosophy given as to why those tools, and in that order, have been chosen. Whilst I am very familiar with these tools, as are obviously the author's, if they would like their suite of tools to be adopted as a standard protocol - or at least a starting point - it might help their case to walk through the more novice users with their design choices in a more friendly format.

Whilst I don't think the issues with the documentation, or lack of walktrhough examples, and missing dataset detract from the paper or the tool in and of itself, for a tool that is described as "easy to use" and aimed at including more researchers being able to produce better phylogenies (and so better test their hypotheses) via a standardised pipeline (especially those who may not have much experience in determining what they need to start with) I would expect there to be much better documentation, and for the database to be accessible immediately. The merit of the paper and the security of it rests on it being tested externally before publication. Not least for it to be considered to be published in a PloS journal. 

Minor corrections to Text:

Ln109 "base" should be "based"

Ln189 insert "a" between "by" and "few"

Ln190 remove "more" between "much" and "widely"

Ln238 insert "that" between "analyses" and "are"

Supplementary

Ln145 script name is missing an "s"

Ln149 "choose" should be "chose"

Ln162/165 script name is incorrect

Ln 173 "the taxa" should be "each taxa"

Ln175 "culled" should be "retrieved"

Ln179 comma after "taxa"

Ln186 mentions inaccessible file

Ln193 "organisms"

Ln200 - where is this tool ParaSorter?!

Ln202 - not shipped?!

Ln211 - this should be subset_tools.py

Ln217 - "no_gaps_stops.py" doesn't exist?

Ln387+388+412 - script name is wrong compared to github

Ln432/3 program name incorrect

Ln560 program name incorrect

---

## [Decision Letter · Decision Letter 1]

13 Jul 2021

Dear Matt,

Thank you for submitting your revised Methods and Resources paper entitled "PhyloFisher: A phylogenomic package for resolving eukaryotic relationships" for publication in PLOS Biology. I've now obtained advice from the original reviewers and have discussed their comments with the Academic Editor. 

Based on the reviews, we will probably accept this manuscript for publication, provided you satisfactorily address the following data and other policy-related requests.

IMPORTANT:

a) Our IThenticate software detects a significant similarity between lines 418-427 and a BMC Biology paper (https://bmcbiol.biomedcentral.com/articles/10.1186/s12915-021-01007-2) by a subset of your co-authors. Please could you adjust the wording of this section to minimise any copyright issues?

b) Please could you attend to my Data Policy requests further down. Essentially, while it may be that some of your Figs can be generates using the software package and the database provided, it's not clear whether this applies to all of the Fig panels (especially the supplementary Figs S2, S3, S4, S5, S6, S7, S8, S10, S12, S13, S14). In each case, please can you supply the underlying data and cite its location clearly in the respective Fig legends? If the data can be generated using the Phylofisher software and the dataset then please specify this (including URLs) in each legend.

We expect to receive your revised manuscript within two weeks. 

*Published Peer Review History*

*Early Version*

Sincerely,

Roli

Senior Editor,

rroberts@plos.org,

PLOS Biology

DATA POLICY:

Regardless of the method selected, please ensure that you provide the individual numerical values that underlie the summary data displayed in the following figure panels as they are essential for readers to assess your analysis and to reproduce it: Figs S2, S3, S4, S5, S6, S7, S8, S10, S12, S13, S14.

IMPORTANT: Please also ensure that figure legends in your manuscript include information on where the underlying data can be found, and ensure your supplemental data file/s has a legend. If the data can be generated using the Phylofisher software and the dataset then please specify this (including URLs) in each legend.

DATA NOT SHOWN?

REVIEWERS' COMMENTS:

Reviewer #1:

The authors have added an additional analysis of another area of the tree of life. This is adding to the applicability of the software. The authors have furthermore addressed most of the comments of the reviewers. The addition of more authors is justified.

Reviewer #2:

They have sufficiently addressed most of the comments and made changes that have strengthened the manuscript. While I don't necessarily agree with all of their responses, it's not imperative that I agree with everything for this to be publishable. This is a good contribution to the community who works on resolving deep relationships in the eukaryotic tree of life.

Reviewer #3:

Initially I was worried that the paper and accompanying tool did not do enough to allow users to get started easily, however this has been massively improved on re-submission. There is a now a very well documented manual that accompanies the set of scripts/programs outlined in the paper. The tools themselves have also been improved and the starting database is now available. These inclusions and updates remove any doubts that it will not be a useful tool for the community. Therefore I am happy to recommend the manuscript's acceptance and the author's replies subsequently I have no further recommendations and am looking forward to using the software in some analyses.

---

## [Editor Report · Decision Letter 2]

15 Jul 2021

Dear Matt,

On behalf of my colleagues and the Academic Editor, Andreas Hejnol, I'm pleased to say that we can in principle offer to publish your Methods and Resources paper "PhyloFisher: A phylogenomic package for resolving eukaryotic relationships" in PLOS Biology, provided you address any remaining formatting and reporting issues. These will be detailed in an email that will follow this letter and that you will usually receive within 2-3 business days, during which time no action is required from you. Please note that we will not be able to formally accept your manuscript and schedule it for publication until you have made the required changes.

IMPORTANT:

a) As discussed by email, please ensure that the data underlying the Figures (conveyed to me via Figshare) is available in the stipulated location (https://ir.library.msstate.edu/...) by the time that the paper publishes.

b) Please mention the location of the data clearly in all relevant main and supplementary Figure legends (e.g. "The data underlying this Figure may be found at https://ir.library.msstate.edu/...."). I'm aware that this will look repetitive, but it makes the Figs+legends more standalone. I'll tell my colleagues to expect this change.

PRESS: We frequently collaborate with press offices. If your institution or institutions have a press office, please notify them about your upcoming paper at this point, to enable them to help maximise its impact. If the press office is planning to promote your findings, we would be grateful if they could coordinate with biologypress@plos.org. If you have not yet opted out of the early version process, we ask that you notify us immediately of any press plans so that we may do so on your behalf.

Sincerely, 

Roli

Roland G Roberts, PhD 

Senior Editor 

PLOS Biology

rroberts@plos.org